# State-of-the-Art Approaches for Image Deconvolution Problems, including Modern Deep Learning Architectures

**DOI:** 10.3390/mi12121558

**Published:** 2021-12-14

**Authors:** Mikhail Makarkin, Daniil Bratashov

**Affiliations:** Biomedical Photoacoustics Lab, Saratov State University, 83 Astrakhanskaya Str., 410012 Saratov, Russia; makmih50@gmail.com

**Keywords:** image processing, deconvolution, deep learning, digital microscopy

## Abstract

In modern digital microscopy, deconvolution methods are widely used to eliminate a number of image defects and increase resolution. In this review, we have divided these methods into classical, deep learning-based, and optimization-based methods. The review describes the major architectures of neural networks, such as convolutional and generative adversarial networks, autoencoders, various forms of recurrent networks, and the attention mechanism used for the deconvolution problem. Special attention is paid to deep learning as the most powerful and flexible modern approach. The review describes the major architectures of neural networks used for the deconvolution problem. We describe the difficulties in their application, such as the discrepancy between the standard loss functions and the visual content and the heterogeneity of the images. Next, we examine how to deal with this by introducing new loss functions, multiscale learning, and prior knowledge of visual content. In conclusion, a review of promising directions and further development of deconvolution methods in microscopy is given.

## 1. Introduction

Progress in modern imaging optics is still limited by the physical limitations of image resolution caused by the wave nature of light. The major fundamental constraint is the diffraction limit, but there are also limitations associated with the individual technical features of devices [1]. Deconvolution can be used to work around these constraints; it is one of the possible approaches for obtaining super-resolution. It allows us to obtain images with a higher resolution than allowed by physical methods [2]. In addition to optical limitations, various distortions are encountered in microscopy. These include scattering (random disturbance of light caused by differences in the sample’s refractive index and its environment), glare (random disturbances caused by the unexpected appearance of a beam of light with inappropriate polarization), and blur. Blur often appears when recording an image of a moving sample or if the camera does not have enough temporal resolution—this is referred to as motion blur [3]. When this sample is displaced from focus, this is referred to as defocusing blur [4]. In addition, blurring appears with a simple shaking of the optical device itself [5] (this is also a kind of motion blur). Modern microscopy methods based on structured illumination (for example, light microscopy [6]) work with complex-shaped light beams that can introduce serious geometric distortions [7], which affect the visible shape of cells and particles in the image. Therefore, they can serve as a good example of the distortion that is inevitable when using an optical system. In addition, due to the beam’s profile, these distortions can have spatial inhomogeneity of—different parts of the image will have different degrees of distortion. However, the advantages of these methods (high acquisition speed with low photodamage, high resolution, and contrast) are enormous. Therefore, it is necessary to use computational methods to unleash their potential without interfering with the physical mechanisms of their work.

It is possible to eradicate artifacts using purely instrumental methods. Special anti-reflective coatings and polarizing filters in modern microscopes make negligible the likelihood of glare and scattering [8]. High-speed acquisition systems and the synchronizing of particle flow with image acquisition (virtual freezing [9]) can reduce problems with motion blur [10]. To solve problems with defocusing, autofocus systems are used in modern designs [11]. However, the use of these tools is not always justified. They are expensive, make optical systems complex, and are often difficult to operate and maintain. At the same time, there are several modern applications, such as the composite cameras of a smartphone or sensor fusion in autonomous driving, which are computational photography technologies, where the true images of objects are restored from a series of low-quality images obtained by a group of sensors with different characteristics. The adaptation of the modern techniques of machine image compositing can significantly simplify the design of optical microscopy systems. These methods are based on the deconvolution of the resulting image. In its mathematical form, the deconvolution of images can be represented through the definition of a distorted image. The distorted image is represented as a convolution of two functions (Figure 1):(1)h∗x=y

Here, x is the original function of the image to be restored, h is the distortion function (in other terminology, blur kernels, the response of the optical system, and point spread function), y is the final image, and ∗ symbol denotes the convolution operation. Usually, it is also necessary to somehow take into account the presence of additive noise:(2)(h∗x)+ε

In actuality, after carrying out the inverse transformation of Expression (2), you can again obtain the original image x. This operation will be the deconvolution of the image. It should be noted that one of the main difficulties arising during the deconvolution operation is the amplification of additive noise. The amount of amplification will significantly depend on the nature of the noise and the method used. For example, we can say that the more restrictions the methods contain (for instance, they can take into account only Gaussian noise), the more they are susceptible to noise amplification and the appearance of artifacts. This problem has led to a shift towards methods that extract the required parameters directly from the provided data.

Deconvolution is widely used in modern microscopy systems, from confocal and structured light illumination techniques [13,14] to specialized systems designed for ophthalmology [15]. Even without using some special techniques, it increases the resolution in optical microscopy several times and significantly improves image quality for detecting objects’ shape and size. These parameters are essential in various areas of biomedical and clinical research.

## 2. Deconvolution Types

The works mentioned above use the well-known point spread function (PSF) of the optical system. The PSF describes the imaging system’s response to the infinitely small object (single point size). The PSF of an imaging system can be measured using the small calibration objects of the known shape or the calculated from the first principles if we know the parameters of the imaging system. Therefore, these and similar methods are classified as non-blind deconvolution methods. However, more often than not, the PSF cannot be accurately calculated for several reasons. First, it is impossible to take into account all the noise and distortions that arise during shooting. Second, the PSF can be very complex in shape. Alternatively, the PSF can change during the experiment [16,17]. Therefore, methods have been developed that extract the estimated PSF directly from the resulting images. These methods can be either iterative (the PSF estimate is obtained from a set of parameters of sequentially obtained images that are refined at each pass of the algorithm) or non-iterative (the PSF is calculated immediately by some parameters and metrics of one image).

The mathematical formulation of the blind deconvolution problem is a very ill-posed problem and can have a large (or infinite) number of solutions [18]. Therefore, it is still necessary to impose certain restrictions on the condition—to introduce regularization, for example, in the form of a so-called penalty block, such as a kernel intensity penalizer [19] or a structured common least norm [20] (STLN), or in other ways. A typical problem, in this case, is the appearance of image artifacts [21,22], which appear due to an insufficiently accurate PSF estimate or because of the nature of the noise. This problem is especially acute for iterative deconvolution methods, since there is a possibility that PSF and noise in different images will not coincide with real ones, and therefore the error accumulates at each iteration (Figure 2). In this regard, machine learning algorithms are of particular interest because they are specifically geared towards extracting information from data and their iterative processing.

An important issue when solving the deconvolution problem is the nature of the distortion. It can be uniform (that is, the same distortion kernel and/or noise is applied to all parts of the image) or non-uniform (different blur kernels are applied to different parts of the image and/or the noise on them is also not the same). The absence of a uniform distortion for the image further complicates the task. In these cases, it is no longer possible to proceed with a general estimate that we derive from large-scale dependencies between pixels. Instead, we have to consider local dependencies in small areas of the image, which makes global dependencies more complex structures in a mathematical sense, and much more expensive from a purely computational perspective. Figure 3 shows an example of non-uniform blur. Accordingly, to different types of distortion must be applied different deconvolution types, either uniform or non-uniform. In the future, we will use as synonyms the concepts of homogeneous and uniform, heterogeneous and non-uniform.

In real-life problems, one must confront the problem of inhomogeneous distortions most often.

## 3. Deconvolution Methods Classification

Modern deconvolution algorithms include linear, nonlinear, statistical, and cluster analysis methods. Additionally, these can be separated into non-blind and blind deconvolution algorithms.

Linear methods include similar algorithms for the inverse filter and Wiener deconvolution, Tikhonov filtering [25], and linear least squares [26]. Take Wiener’s deconvolution as an example [27]. The observed image can be expressed as:(3)y=(h∗x)+ε

In this formulation, y is the observed image, x is the undistorted image, h is the linear time-independent response of the system (introducing distortions), and ε is the unknown additive noise that is independent of x. A rough estimate of the undistorted image will look like this:(4)$$\hat{x}=\left(g*y\right)$$
where g is the new deconvolution kernel. The main idea of the method is to use not the functions themselves, but their Fourier transforms. With their help, it will be possible to obtain the inverse operator (based on the convolution theorem). Capital letters correspond to the Fourier transforms of the corresponding functions. The formula is as follows:(5)$$G\left(f\right)=\frac{H\left(f\right)S\left(f\right)}{{\left|H\left(f\right)\right|}^{2}S\left(f\right)+N\left(f\right)}$$
and
(6)$$G\left(f\right)=\frac{1}{H\left(f\right)}\left[\frac{1}{1+1/({\left|H\left(f\right)\right|}^{2}S\left(f\right)SNR\left(f\right))}\right]$$
where $S\left(f\right)=E{\left|X\left(f\right)\right|}^{2}$ and $N\left(f\right)=E{\left|V\left(f\right)\right|}^{2}$ are the average spectral powers of x and ε, respectively.

Since linear methods are inferior at handling complex noise and image representation functions, various forms for introducing nonlinearity are provided by the nonlinear methods. This group of methods includes the classic Richardson–Lucy algorithm [28], Janson van Cittert [29], a nonlinear least-squares method [30], and the iterative Tikhonov–Miller method with regularization [31].

Statistical methods are based on various methods for estimating parameters from mathematical statistics. These include methods based on a maximum likelihood estimates or an estimate of the posterior maximum [32,33,34]. Many deconvolution algorithms are either entirely based on statistical methods or use many of their elements to one degree or another.

Cluster analysis methods can be roughly represented as methods that divide specific pixels of the image according to the degree of similarity with their neighboring pixels and further grouping them according to the resulting classes. This includes the nearest-neighbor, k-nearest-neighbor, and k-weighted nearest-neighbor methods [35,36]. Today this group of methods is very rarely used.

Methods from each group can be either iterative or non-iterative. As a rule, iterative methods of computational deconvolution find broader applications in practice since they allow one to achieve better results, while the general trend towards an increase in computing power means that their hardware requirements are no longer a serious issue.

To one degree or another, iterative algorithms often include statistical methods for evaluating the resulting image parameters, which are refined with each new pass of the algorithm [37]. A classic example of an iterative algorithm is the modified Richardson–Lucy algorithm [15,28]. With non-iterative methods, a strict definition of some metric is often required, from which the algorithm will build on in its work. For example, the APEX algorithm [38] requires a rough idea of the PSF shape, while the SeDDaRA algorithm [39] requires a reference scene image. In ref. [40], Tikhonov’s regularization was used with the Sobolev smoothing operator under periodic boundary conditions (this method was extended in [41]). Since the modern problems facing optics, computer vision, and astronomy are often blind deconvolution problems, the review will focus on the corresponding methods.

Since, in the case of heterogeneous distortion, the relations between local regions will most often be nonlinear, classical methods have limited possibilities for their deconvolution. Therefore, one has to independently build complex mathematical models [42], which can be transferred only poorly to other cases. However, the use of machine learning can solve this problem.

Machine learning algorithms find common patterns in the training dataset (in our case, a set of images) and predict their appearance in new data. The algorithm will receive a training set of images as an input and iteratively find spatial patterns in them. They will encode the true PSF or a set of features typical of distorted or undistorted images directly. Then, the operation of the algorithm is checked on a test set. If the results are satisfactory, then the model reconstructs images from natural experimental data. They can also be used with non-blind cases (e.g., see [43,44,45]) and blind deconvolution [46,47].

Another advantage of using computational deconvolution methods is that they are weakly domain specific. Of course, there are certain peculiarities of the data used in microscopy and imaging cytometry. However, in general, it can be argued that algorithms that show good results in other areas can be used in this area without major modifications. For example, the Richardson–Lucy algorithm is widely used in microscopy [48,49]. Thus, it successfully handles both the case of structured illumination and the case of three-dimensional image reconstruction. Use of the van Cittert algorithm and Wiener filter can be seen in [50]. Statistical [51,52] and various nonlinear methods [53,54] are also used.

A striking example of the power of computational deconvolution methods can be found in their application to microscopy. They have given rise to significant progress in obtaining ultra-high resolution and with cleaning the image of artifacts and distortion. In article [55], which uses frequency analysis and expansion according to the Gaussian function, the possible limits of the algorithm for maximizing the mathematical expectation are indicated. It was shown that, theoretically, it could be used to distinguish objects that are eight times smaller than the diffraction limit that the used optical system allows. On the one hand, it was possible to achieve super-resolution by purely computational methods. On the other hand, these methods have disadvantages; the main one is the high complexity of computations, which requires the use of high-performance systems. Today, the parallel improvement of physical and computational methods makes it possible to distinguish objects that are orders of magnitude smaller, including those tens of nanometers across.

## 4. Application of Deep Learning in a Deconvolution Problem

Machine learning is divided into classical (ML) and deep learning (DL). Classical algorithms are based on manual feature selection or construction. Deep learning transfers the task of feature construction entirely to the neural network. This approach allows the process to be fully automated, and performs blind deconvolution in the complete sense, i.e., restoring images only using information from the initial dataset. Therefore, the solution to the deconvolution problem using DL is an auspicious direction at the moment. The automation of feature extraction allows these algorithms to be adapted to the variety of resulting images, which is crucial since it is almost impossible to obtain an accurate PSF estimate and reconstruct an image by using it in the presence of random noise and/or several types of parameterized noise. A more reasonable solution would be to build its iterative approximation, which will adjust when the input data changes, as DL does. Today, two main neural network types are used for deconvolution.

The first is convolutional neural networks (CNN). In CNN, alternating convolutional and downsampling layers extract from the image a set of spatially invariant hierarchical features, a set of low-level geometric shapes, and transformations of pixels that line up into specific high-level features [56]. In theory, in the presence of “blurred/non-blurred image” pairs, CNN can learn a specific set of transformations for image pixels that lead to blur (i.e., evaluate the PSF) (Figure 4 and Figure 5). For example, in [57], the authors show that, on the one hand, such considerations are relevant; on the other hand, they do not work well for standard CNN architectures and do not always produce a sharp image. The reason is that small kernels are used for convolutions. Because of this, the network is unable to find correlations between far-apart pixels.

Nevertheless, using CNNs for image restoration problems leads to the appearance of artifacts. The simple replacement of small kernels with large kernels in convolutions does not generally allow the network to be trained due to the explosion of gradients. Therefore, the authors replaced the standard convolutions with the pseudoinverse kernel of the discrete Fourier transform function. This kernel is chosen so that it can be decomposed into a small number of one-dimensional filters. Standard Wiener deconvolution is used for initial activation, which improves the restored image’s sharpness.

However, the convolution problem is not the only one. When using the classic CNN architectures (AlexNet [58], VGG [59]), researchers have found that they perform poorly at reconstructing images with non-uniform backgrounds, often leaving certain areas of the image blurry. One can often find such a phenomenon as the fluctuation of blur during training—under the same conditions and on the same data, with frozen weights, the network after training still gives both sharp and blurry images. What seems paradoxical is that an increase in the training sample number and an increase in the depth of the model led to the fact that the network began to recover images with blur more often. This is due to some properties of the CNN architectures, primarily the use of standard loss functions. As shown in [60,61], blur primarily suppresses high-frequency areas in the image (which means that the L-norms of the images decrease). This means that with the standard maximum a posteriori approach (MAP) with an error function that minimizes the L_1_ or L_2_ norm, the optimum of these functions will correspond to a blurry image, not a sharp one. As a result, the network learns to produce blurry images. Some modification of the regularization can partially suppress this effect, but this is not a reliable solution. In addition, the estimation for minimizing the distance between the true and blurred image is inconvenient because, if there are strongly and weakly blurred images in the sample, neural networks are trained to display intermediate values of blur parameters. Thus, they either underestimate or overestimate blur [62]. Therefore, using CNN for the deconvolution task requires specific additional steps.

The use of multiscale learning looks promising in this regard (Figure 6). To obtain a clean image after deconvolution, we need to solve two problems. First, find local patterns on small patches so that small details can be restored. Second, consider the interaction between far-apart pixels to capture the distortion pattern typical to the image. This requires the network to extract spatial features from multiple image scales. It also helps to learn how these traits will change as the resolution changes. In ref. [63], the authors propose using a neural network architecture called CRCNN (concatenated residual convolutional neural network). In this approach, residual blocks are used as the elements of spatial feature extraction in an implicit form, and are then fed into an iterative deconvolution (IRD) algorithm. They are then concatenated at the output to obtain multiscale deconvolution. In addition, the approach described in [64] integrates the encoder–decoder architecture (see, for example, [65]) and recurrent blocks. A distorted image at different scales is fed into the input of the network. When training a network, the weights from the network’s branches for smaller scales are reused, with the help of the residual connection when training branches for larger ones. This reduces the number of parameters and makes learning easier. Another important advantage of multiscale learning is the ability to completely abandon the kernel assessment and end-to-end modeling of a clear image. The general idea [66] is that co-learning the network at different scales and establishing a connection between them using modified residual blocks allows a fully fledged regression to be carried out. We are not looking for the blur kernel, but approximating a clear image in spatial terms (for example, the intensity of the pixels at a specific location in an image). At the moment, the direction of using multiscale training looks promising, and other exciting results have already been obtained in [67,68,69]. We can separately note an attempt to use the attention mechanism to study the relationship between spatial objects and the channels on an image [70].

The second type of architecture used is generative models, which primarily includes various modifications of generative adversarial networks (GANs) [71]. Generative models try to express a pure latent image explicitly. Information about it is implicitly contained in the function space (Figure 7). GAN training is closely related to the previous issue discussed above: prioritization and the related training adjustments. The work [72] used two pre-trained generative models to create non-blurred images and synthetic blur kernels. Next, grids were used to approximate the real kernel of the blur using a third, untrained generator. In ref. [73], a special class network—spatially constrained generative adversarial network (SCGAN) [74]—was used, which can directly isolate spatial features in the latent space and manipulate them directly. This feature made it possible to modify it for training on sets of images projected along three axes, implementing their joint deconvolution, and obtaining a sharp three-dimensional image. When using GAN, the problem of the appearance of image artifacts almost always plays a unique role. At the beginning of the training cycle, the network has to make strong assumptions about the nature of the noise (for instance, Gaussian, Poisson) and its uniform distribution across all of the images. The article [75] proposes eradicating artifacts without resorting to additional a priori constraints or additional processing. The authors set up a convolutional network as a generator and trained it to produce sharp images with its error function independently. At the same time, there remained a common error function for the entire GAN, which affected setting the minimax optimization. A simplified VGG was taken as a generator, which determined whether the input image was real or not. As a result, CNN and GAN worked together. The generator updated its internal variable directly to reduce the error between *x* and G(z). The GAN then updated the internal variable to produce a more realistic output.

The general problems affecting blind deconvolution are still present while using a DL approach, albeit with their own specificities. Blind deconvolution is an inverse problem that still requires sufficiently strong prior constraints, explicit or implicit, to work (Figure 8). As an example of explicit constraints, one can cite the above assumptions about the homogeneity of noise in images, the assumption that optical aberrations are described by Zernike polynomials [76], or directly through special regularizing terms [77]. A good example of implicit constraints is the pretraining generator networks in a GAN or the training discriminator networks that use certain blurry/sharp images sets. These actions automatically determine the specific distribution in the response space corresponding to the training data. According to this distribution, control signals are generated and supplied to the generator. It will adjust to this distribution and produce an appropriate set of synthetic PSFs or parameters for a clean image. This approach allows extracting prior constraints directly from the data. This property is typical for generative models in general; for example, the combination of an asymmetric autoencoder with end-to-end connections and a fully connected neural network (FCN) is used in [78]. The autoencoder creates a latent representation of the clean image, and the FCN learns to extract the blur kernels from the noise and serves as an additional regularizer for the autoencoder. The coordinated interaction of these networks makes it possible to reduce the deconvolution problem, contributing to a MAP optimization of the network parameters.

The clear advantages of the neural network approach include the already mentioned full automation, the ability to use the end-to-end pipeline (which significantly simplifies the operation and debugging of the method, or its modification if necessary), and its high accuracy. The disadvantages include problems common to DL—the need for sufficiently large and diverse datasets for training and computational complexity (especially for modern CNN and GAN architectures). However, it is worth highlighting the weak interpretability of the results—even with the deconvolution problem, it is often necessary to restore the image, estimate the PSF, and understand how it was obtained.

In addition to GAN and CNN, other architectures are used for the deconvolution problem. An autoencoder consists of two connected neural networks—an encoder and a decoder. The encoder takes input data and transforms it, making the representation more compact and concise. In the generating subtype, the variational autoencoder, the encoder part produces not one vector of hidden states but two vectors—mean values and standard deviations—according to which the data will be restored from random values. In addition to [78], the article [79] can be noted. It uses the output of a denoising autoencoder, which is typically the local average of the true density of the natural image, while the error of the autoencoder is the vector of the mean shift. With a known degradation, it is possible to iteratively decrease the value of the average shift and bring the solution closer to the average value, which is supposed to be cleaned of distortions. In the article [80], an autoencoder was used to find invariants on a clean/blurry image, based on which the GAN was trained to restore clean images. Autoencoders are commonly used to remove noise by transforming and compressing data [81,82,83].

For the case of video (when there is a sequential set of slightly differing images), some type of recurrent neural network (RNN) is often used, most often in combination with CNN. In work [84], individual CNNs first obtained the pixel weights of the incoming images from the dynamic scene and extracted its features. The four RNNs then processed each performance map (one for each direction of travel), and then the result was combined by the final convolutional network. This helped increase the receptive field and ensured that spatial non-uniformity of the blur was taken into account. In work [85], based on the convLSTM blocks, a pyramidal model of the interpolation of blurred images was built, which provided continuous interpolation of the intermediate frames, building an averaged sharp frame, and distributing information about it to all modules of the pyramid. This created an iterative de-blurring process. An interesting approach is proposed in [86], which offers an alternative to multiscale learning, called multi-temporal learning. It does not restore a clean image at small scales then go to the original resolution, but rather works in a temporal resolution. A strong blur is a set of weak blurs that are successive over time. With the help of the RNN, the correction of weak initial blurring is iterated over the entire time scale.

Interest in the use of attention mechanisms in the task of deconvolution is beginning to grow. Attention in neural networks allows one to concentrate the processing of incoming information on its most important parts and establish a certain hierarchy of relations between objects to each other (initially, attention mechanisms were developed in the field of natural language processing and helped to determine the context of the words used). Attention mechanisms can be implemented as separate blocks in classical architectures and used, for example, to store and further summarize global information from different channels [87] or to combine hierarchical functions from different points in time in a video in a similar way [88]. Attempts are being made to use an architecture entirely based on attention—the so-called transformers [89]. An illustrative example of their use is shown in [90]. The authors take advantage of one of the main advantages of the transformer—the ability to handle global dependencies. Using residual links and building the architecture in a similar Y-net style allows the user to obtain local dependencies and link them to larger ones. As mentioned above regarding multiscale learning, this is one of the main problems in image deconvolution and, perhaps, the attention mechanism will allow one to solve it more efficiently—in some tasks (for example, removing rain and moiré patterns), the increase in PSNR and SSIM is very noticeable.

Deep learning is also beginning to be widely used in microscopy and related techniques. Yang et al. showed an example of non-blind deconvolution by using neural networks for 3D microscopy with different viewing angles for samples [91]. Usually, CNNs have some difficulty recognizing rotations, so the authors used GANs with a self-controlled approach to learning. It allowed them to surpass the standard algorithms for three-dimensional microscopy (CBIF [92], EBMD [13]) in terms of quantitative and qualitative characteristics: PSNR (peak signal–noise ratio), SSIM (structural similarity index), and CC (correlation coefficients). The flexibility of deep learning can also be seen in [93], which shows how it can be combined with the classical method. In this work, adaptive filters were used with the Wiener—Kolmogorov algorithm. The neural network predicted the optimal values of the regularizer and adjusted the filter kernel for it. It helped to improve the quality of the resulting image. In this case, the computational time was less than with direct image restoration using neural network training. It is an important point—in microscopy, especially in imaging cytometry, a transition to real-time image processing is needed. Examples can be given where super-resolution is achieved using the GAN [94,95], the encoder–decoder [96], and the U-Net-based architectures [97]. The use of deep learning in these works made it possible to significantly improve the quality of the reconstructed image and remove the connection to the optical properties of the installations, reducing the problem to a purely computational one. Despite the demanding computational power of deep learning algorithms, against the background of classical methods (especially nonlinear ones), they can show excellent performance [98] precisely due to their ability to build hierarchical features. Their other feature is the need for large datasets for training. On the one hand, the need is satiated by the appearance of large databases of cellular images in the public domain; on the other hand, it is still a problem. However, in imaging flow cytometry, a large dataset is relatively easy to collect. Therefore, it is convenient to use neural networks in flow cytometry with visualization; for example, the residual dense network can be used to eradicate blurring [99].

The use of deep learning made it possible to cope with non-uniform distortion, which classical methods could hardly achieve, and therefore weakened the requirements for the quality of the restored images. In addition to increasing the numerical characteristics, deep learning allows one to automate the image recovery process thoroughly and, therefore, expand its use for non-specialist users.

Furthermore, deep learning is used in medical imaging, by which we mean here all non-microscopic instruments (e.g., CT, MRI, and ultrasound). In ref. [100], a standard approach was used to extract the main features of images with a convolutional network at a low resolution and restore a pure image at a higher resolution. A similar but slightly complicated method is shown in [101], where the so-called upsampling layer was additionally used in the network architecture (the terminology requires caution: the layer was previously called the deconvolution layer, but it did not perform the deconvolution operation and this caused confusion). The methods were tested on MRI and retinal images. Chen et al. proposed adding residual bandwidth to the U-net encoder/decoder network to improve the quality of low dose CT scans [102]. The encoder branch reduces noise and artifacts, and the decoder recovers structural information in the CT image. In addition, a residual gap-filling mechanism will complement the details lost when going through multiple layers of convolution and deconvolution. Finally, in [103], the convolutional network was trained to recognize the difference between low- and high-resolution MRI heart slices, and then taught to use this information as a recovery operator.

Deep learning is being used to improve image quality in photography and video. In addition to the usual adaptation of algorithms to, for example, mobile devices (see the review [104]), it can also be found being applied to the restoration of old photographs [105] or shooting fast-moving objects [106].

Moreover, deep learning deconvolution is used in astronomy [107,108,109]. In this area, observations are inevitably prone to distortion due to the presence of the atmosphere. In addition, they are often inhomogeneous due to the turbulence of the air masses, so the use of DL helps a lot.

## 5. Features of Training, Testing, and Validation in Deep Learning

Engineering plays an important role in deep learning, especially with data collection and preparation, model training, and validation. However, we will not dwell on the general pipelines in detail, and instead we will consider the features associated with solving the deconvolution problem.

As mentioned above, neural networks usually require many data. Therefore, any deep neural network is a nonlinear algorithm with a high possible variance for the parameters in the input data. Therefore, drawing up a representative sample for its training will require a significant number of examples. This is doubly true for solving the problem of blind deconvolution because it is an ill-posed problem. Without an explicit transformation of pixels in the image (blur kernels, PSFs), the network will be much more difficult to optimize during training. This will require either an increase in the training time and an adjustment of the hyperparameters, a complication of the model to increase its expressive ability, or an increase in the training sample to increase the statistical significance of the required pattern in the data. The latter method will be, in a sense, the most reliable since it uses the fundamental properties of machine learning algorithms.

At the moment, there are a fairly large number of datasets in the public domain that focus on the problem of removing blur from images. However, the limited amount of data is still a problem. Some of these datasets are too small [60,110]. Some are sharp images that are distorted by synthetic PSFs [66,111,112]. This makes them not quite suitable for real distortion cases. On this side, efforts are already being made to rectify the situation [113]. Furthermore, a significant problem is a lack of datasets for specific areas, for instance, in microscopy and medical imaging.

As a rule, distorted/undistorted image pairs are used for training.

It is important to note some of the features associated with the training stage in DL. A commonplace feature is that GANs (and models with their inclusion) are more difficult to train than models based on convolutional networks. Typical problems (collapsing modes, training stability) are critical, specifically for image deconvolution. Mode collapse is a phenomenon in which the output space of the generator becomes noticeably smaller than the space of the original images. In other words, the generator learns to produce a narrow set of images that the discriminator recognizes as plausible. In this case, the error gradient quickly begins to tend to zero, and the generator output seems to “freeze”, always giving answers from the resulting distribution. The training instability occurs due to the minimax nature of the cost function optimization, which can lead to the fact that the sought-after functions of the discriminator and generator may not converge at all. In real-life problems, various image distortions are often encountered, even in one sample, which creates the danger that the generator will adjust to some specific ones. The difference between this and overfitting is that the model will not necessarily show bad results with new data; instead, we can say that it will be sharpened under one of the distributions encountered and that that part of the images will be restored poorly. There is no general solution to these problems, but there are many tricks that allow one to solve them in specific cases. For example, using the Wasserstein metric inside the error function [12] or the introduction above of a prior generator [72] excludes mode collapse.

As for the testing and validation of DL models, it can be noted that when solving real deconvolution problems, it is better not to use the standard deferred data scheme, i.e., randomly separating from the dataset 80% of examples for a training set and 20% of examples for a test set (or 80% for a training set, 10% for a test set, 10% for a validation set). This is again due to noticeable variance in the data and the possibility of encountering non-uniform blur patterns. Therefore, it is better to use cross-validation, for example, on k-folds or Live-P-Out, even if this creates a large computational load.

Since now we can expect the widespread use of architectures with attention in deconvolution, it is necessary to mention them as well. In a way, they were not immediately used to working with the images served them. Initially, these architectures were very heavyweight, so the search for their optimization immediately began. They were at least partially successful, because using tricks, such as non-overlapping and cyclically shifting attention windows (see [114]), can significantly reduce the number of parameters used by the network and hence the requirements for video memory. With this parameter, networks using transformers can already be compared with CNN.

## 6. Optimization-Based Deconvolution Methods

This group of methods concentrates on a more “mathematical” solution to the deconvolution problem. These methods are (mostly) convex optimization techniques applied to various structures in a sparse matrix representation (Figure 9). They were applied, first of all, to atoms and the atomic norm (the mathematical basis is summarized, for example, in [115]).

The deconvolution problem can be represented as a search for a non-trivial solution to a system of equations. It is the matrix of the observed signals, multiplied by the system’s inverse linear response filter, and will be equal to the undistorted signal. The columns in the undistorted signal matrix are represented as sparse linear combinations of atoms. Non-trivial solutions will be found for the minimum possible number of these atoms. We can reformulate the deconvolution problem in terms of the optimization problem [116].

This approach is better suited for single-image deconvolution. However, it has a significant drawback. In ref. [117], one can see the complexity of using this method. It is based on looking out for similar patches on the image and adding them to dictionaries (Figure 10). GSR (group sparse representation) constraints are imposed on similar patches based on images and kernels to ensure the sparsity of the intermediate latent images and kernels, and L_0_ regularization is added. It is important to note that in the study of sparse data representations, L_0_-regularization is understood not as a regularization based on a mathematically correct L_0_-norm, but as the number of non-zero elements. Although the optimization problem becomes non-smooth and NP complex when using such a regularizer, it allows one to find similar elements accurately. The use of approximate methods fully admits the use of L_0_-regularization [118]. These actions form an understanding of the structure of the image. Then, a multistage optimization is carried out for the image atoms in the dictionaries. The result is a latent representation of a clear image. The PSF is obtained from it and the distorted image. Next, they simply use the standard non-blind deconvolution algorithm and restore the rest of the images. Despite the promising results, the method has problems with non-uniform noise distributions.

**Figure 9 micromachines-12-01558-f009:**
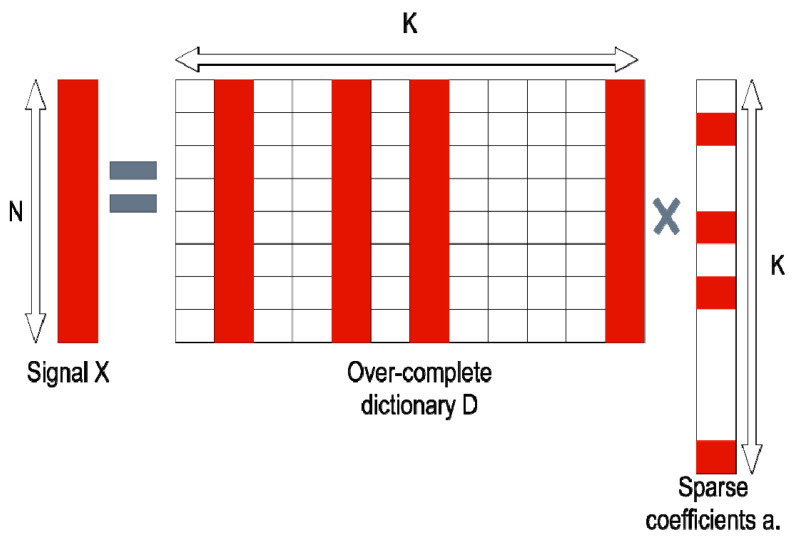
Signal sparse representation (taken from [119]). Dictionary columns are atoms. The signal consists of patches. We think of the signal as an overly complete dictionary of patches multiplied by the sparse coefficients a. N and K are vector lengths of signal X and sparse coefficients a, respectively.

Most importantly, the forced presence of non-smooth regularizers leads to computational complexity. The authors were forced to resort to various tricks (such as approximating an L_0_-regularizer using an L_1_-regularizer). A similar problem can be seen in [120]. The authors used spatially adaptive sparse representation (SASR) as a prior constraint. Blind deconvolution of a single image was carried out with it, and blurring was removed. Then, extended iteration was used in combination with a fast Fourier transform function to solve the joint minimization problem. Despite a different approach to using sparse representation, the same complexities appeared—non-smooth regularization terms.

In other words, very often, a more accurate approximation of the process being optimized will be a non-convex function. At the moment, one can observe how researchers are moving away from convex optimization or fitting for it by examining related problems. An example is the work on multichannel deconvolution [121,122].

Besides atomic optimization, there are other optimization-based deconvolution methods, such as methods involving gradient statistics. They are widely used as standard methods, or as a mixture of Gaussian distributions [123,124] or Laplacians [125] with hyper-Laplacians [126], and specially designed, as in article [127], where a local maximum gradient prior was used. The author’s idea is that blurry images should have fewer gradient values than sharp ones. In work [128], the image is represented as a super-Gaussian field with adaptive structures. This makes it possible to eliminate the inherent problem of the lack of correlation between pixels distant from each other. The field itself is built using high-order Markov random fields, which will integrate the potential of small areas of pixels into a joint probability distribution.

A common disadvantage of all optimization-based methods is that it is rarely possible to create an end-to-end pipeline. Additional actions are always required (forward/backward Fourier transforms, or applying classical deconvolution algorithms at the end). This complicates their application. In addition, the need to carry out many operations with sparse matrices leads to significant computational capacity. The advantages of these methods include a good ability to restore edges.

Optimization methods also find applications in microscopy, for example [129,130,131,132,133]. Their shortcomings are not so critical for this area, since the optical schemes correspond much more strictly to the concept of a linear spatially invariant system. This is much easier to achieve in a laboratory setting. Thus, the convex optimization heuristic does not encounter any particular obstacles in its work. However, this can no longer be said about structured illumination systems. Deep learning is still a more adaptive method.

Since there are situations in which it is impossible to collect data in astronomy, and it is necessary to carry out deconvolution on one image, optimization methods are used there [134,135,136].

## 7. Tools and Instruments

Some differences between the tools for implementing the two modern types of deconvolution algorithms can also be distinguished. First, deep learning-based techniques use a more standardized technology stack. The explosive development of the entire field of deep learning and the growing demand for its application from the business side quite early on required introducing a common denominator for development tools. 

In the early stages of the boom, one could still see the variability in the languages and libraries used. Algorithms could be written entirely in pure C/C++ or use special libraries that are written in it (OpenNN, CNTK), use libraries in Java (Deeplearning4j) or a specialized computer algebra system (Mathematica, MATLAB), or a combination of a compiled/scripting language for low-level implementation and an interface (Torch, Caffe [137]). The latter is currently the most widespread. The use of a (usually statically) compiled language allows for heavy optimization computations; a scripting language (more often with dynamic typing) as an interface facilitates rapid prototyping of the model. 

Today, as a rule, one of two deep learning libraries is used—Tensorflow [138] (supported by Google) or Pytorch (supported by Facebook). It can be argued that they represent a kind of standard. Both are written using the C++, Python, and CUDA languages, and have a special C-like interface for efficient parallel programming on video cards from NVIDIA. This capability is significant for deep learning applications. In Pytorch, however, there are elements written in pure C and Fortran. The interface of both libraries can use several languages, but most often it is Python. Both Tensorflow and Pytorch represent computation as an automatically differentiable dynamic graph. Not so long ago, Tensorflow used a static graph, but this feature was less convenient.

In optimization-based methods, there is no such unity of the tools used. As a rule, algorithms are implemented in C, C++ (since high computational speed is critical), or MATLAB. However, specialized libraries have not yet appeared due to the noticeably lower prevalence of these methods.

Table 1 below shows the specialized deep learning software tools that are common today.

As for the hardware, most computing uses GPUs from NVIDIA, in a desktop or cloud version. Graphics processors were initially designed for the efficient parallel computation of many vector algebra operations and are well suited for deep learning. Recently, even more specialized devices have appeared, such as a tensor processor from Google, but they are not yet common in the open market and are either prototypes or only available through the cloud.

## 8. Discussion

Modern neural network architectures require quite significant computational resources for their operation. This is an obstacle to their widespread use. This problem is solved through the optimization of the architecture. The results of training a neural network can be represented as sparse data representations, the specific configuration of which is highly dependent on the specific hyperparameters used during training [139]. It can be argued that most weights and ratios between them will not be used at some point in the workout. Therefore, neural networks have a huge reserve for “reduction”. This is confirmed by the recent emergence of a multitude of so-called “lightweight architectures”, which are not that inferior in accuracy compared to complex ones, but which are much simpler and require less computational resources [140,141,142]. Most likely, we can expect the same to appear in architectures for deconvolution. In many areas of its application, it is desirable to switch to real-time mode (or at least obtain the minimum delay possible) either for devices with limited computational power (for example, smartphones) or with sufficient power, but with a huge data volume (for example, in the imaging flow cytometry [143]). For example, modern microscopy and cytometer devices are based on smartphone cameras [144], which makes it possible to provide low-cost microscopy facilities for the field research.

A more theoretical direction of research is the search for a better mathematical representation of the concept of “image sharpness” and its assessment. It has already been said above that the standard error functions for neural networks do not quite adequately describe the desired result. At present, intensive research work is already underway in this area. For example, in [145], in their two-phase pipeline, the authors proceeded from the assumption that human perception in recognizing blurry and clean images concentrates on the sharpness of the edges, and in [146] they used maps for comparing images of special points in a higher dimension. The similarity of the values of the points in high-order dimensions indicates the similarity of the images. Based on these considerations, the so-called continuous loss is constructed, with the help of which the images closest to sharp are selected. In general, we can talk about a significant field of work in this area. It is not entirely clear what form the objective functions should take in the deconvolution problem.

Optimization-based techniques are now used in certain areas where the use of neural networks is difficult, for example, in rendering for 3D scanning data [147]. Currently, these techniques are used directly in conjunction with deep learning to deconvolve images.

For example, in [148] the problem of the nonlocality of the attention mechanism was solved using a sparse representation. Combining these methods can give noticeable practical results (increase the learning rate, provide convergence with non-convex optimization); therefore, further developments in this area of research can be expected.

Soon, we should expect a purely engineering-focused and rather simple, but at the same time significant step—the unification of datasets and metrics for verification. Table 2 summarizes the most generic described methods. They use standard metrics for the peak signal-to-noise ratio (PSNR) and the structural similarity index (SSIM) and work well with publicly available open datasets.

Summing up the examples given in Section 5 and Section 6, we can safely say that, today, practically no visualization area can move forward without modern computational deconvolution methods. This is most clearly seen in consumer video and photo equipment (including smartphones cameras). Deconvolution and super-resolution techniques have made it possible to reduce the requirements for the optical systems used (and the user’s skills) to a certain extent. Now any user can take high-definition photos and videos without access to professional equipment.

In knowledge-intensive fields, using these methods, especially DL, also made a significant difference in the situation. For example, researchers could go beyond the diffraction limit and obtain clear images under non-uniform blurring conditions. This has seriously increased the potential of optical imaging methods—microscopy and flow cytometry with imaging—and therefore will expand the possibilities of studying biological objects in vivo. For example, today, it is already much easier to observe dynamic changes in single cells using the same flow cytometry with visualization.

Modern deconvolution methods can have the same significant impact on medical imaging, but DL methods are used much less frequently in this area. The reason lies not so much in some technical problems, but rather in strict control in this area. However, the existing examples of use tell us about broad prospects—the possibility of using low-resolution tomographic scans (and with a lower proportion of irradiation) is worth a lot.

In astronomy, the use of deconvolution on optimization methods can still be found as often as DL. Nevertheless, they also produce very clean images without a significant dataset. Astronomy is characterized by the use of adaptive optics and various inhomogeneous effects in images due to the presence of the atmosphere. Since modern methods are better at dealing with blind deconvolution, they are especially valuable for astronomy. In particular, they allow one to eliminate the problem of the fusion of distant light sources due to better denoising.

In the end, it should be added that the automatic DL approach can successfully create software that can be used by scientists with minimal knowledge in the field of computational image restoration, such as biologists, ecologists, doctors, and astronomers. Thus, the time for routine procedures of initial data processing can be reduced.

## 9. Conclusions

Today, it is safe to say that deep learning has become the mainstream approach to image deconvolution. This has allowed researchers to both significantly increase purely quantitative characteristics (for instance, spatial resolution and PSNR) and solve previously inaccessible problems, such as working with spatially inhomogeneous PSF and noise, including blind deconvolution. One way or another, the disadvantages of deep learning can be largely compensated for by various engineering tricks or the accumulation of a sufficient amount of data. Furthermore, the strengths of deep learning (such as adaptability, power, and automatism) far outweigh the disadvantages at this point. Unfortunately, there are not many applications left for optimization-based methods.

The development of new approaches to deconvolution includes new architectures, new techniques, new forms of loss functions, and the collection of huge datasets for training, testing, and validation. An important aspect of modern deep learning for deconvolution is the transfer of learning to low complexity/low power architectures with little loss in accuracy metrics but huge reductions in computational requirements.

However, there is still considerable room for further work. The demand for the resources of modern, powerful neural network architectures, the lack of a consensus on the form of the loss function, and the use of new techniques, such as attention, require further research.

## Figures and Tables

**Figure 1 micromachines-12-01558-f001:**
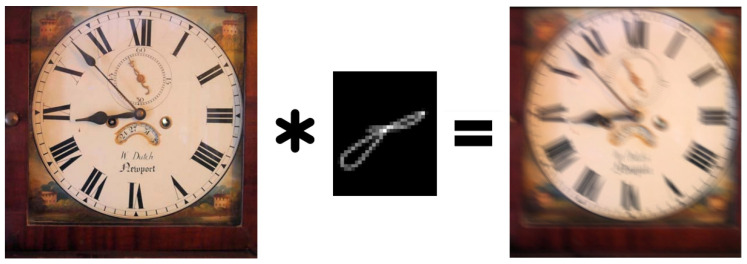
Convolution example. The original undistorted image is shown on the (**left**). In the (**center**), there is the convolution kernel of complex-shaped motion blur. The resulting blurred image is shown on the (**right**). All images are taken from [12].

**Figure 2 micromachines-12-01558-f002:**
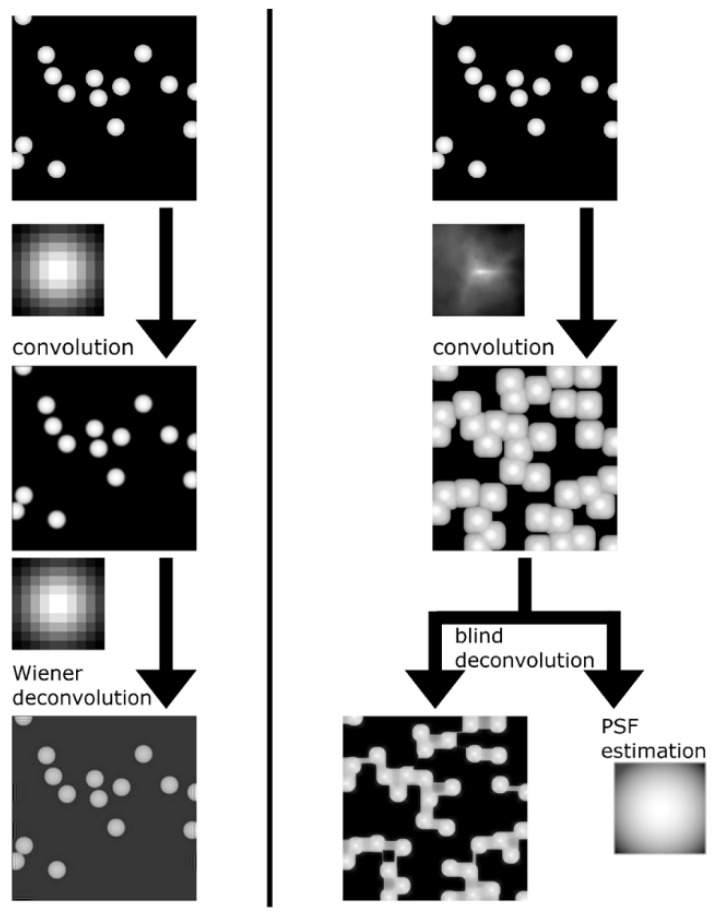
Difference between blind and non-blind deconvolution. On the left is an example of convolution and non-blind deconvolution with the same kernel and some regularization. On the right is an example of poor estimation of PSF with blind deconvolution. A generated image of the same- shaped objects similar to an atomic force microscopy (AFM) image is used as the source. A real AFM tip shape is used as a convolution kernel in this case. Blind deconvolution based on the Villarrubia algorithm [23] confuses object shape and tip shape, resulting in poor image restoration.

**Figure 3 micromachines-12-01558-f003:**
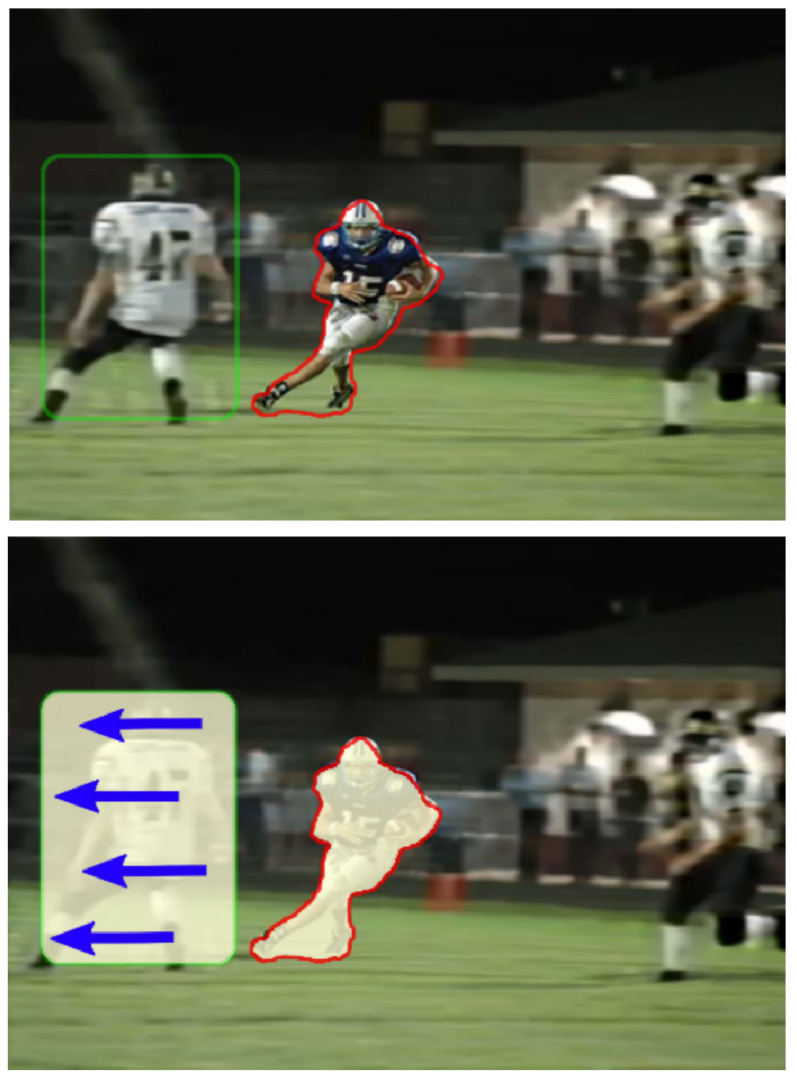
An example of non-uniform distortion. Image is taken from [24]. The operator tracks and focuses on the player with the ball, so there is no blur for his image and the small area around him (red outline). The area indicated by the green outline will show slight defocusing distortion and motion blur (direction of movement is shown by arrows). To adequately restore such an image, it will be necessary to establish the relationships between such areas and consider the transitions between them.

**Figure 4 micromachines-12-01558-f004:**
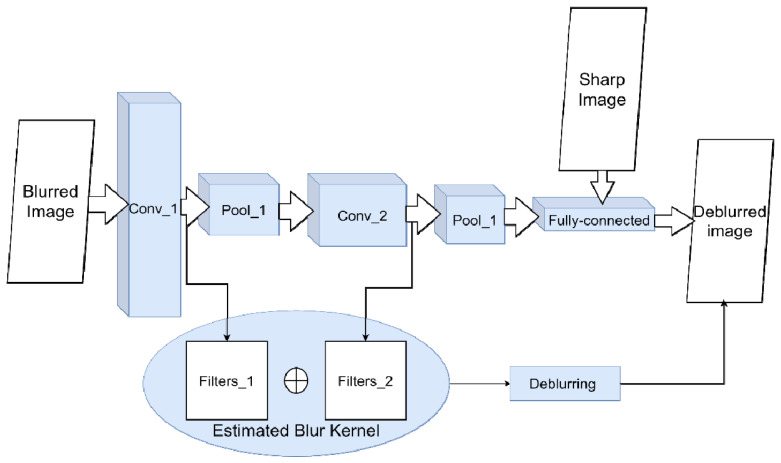
Scheme for image deconvolution with kernel blur using CNN. To solve the classification problem of a blurred/non-blurred image, the neural network adjusts its weights and filters in convolutional layers during training. The sequence of applied filters will be approximately equivalent to the blur kernel.

**Figure 5 micromachines-12-01558-f005:**
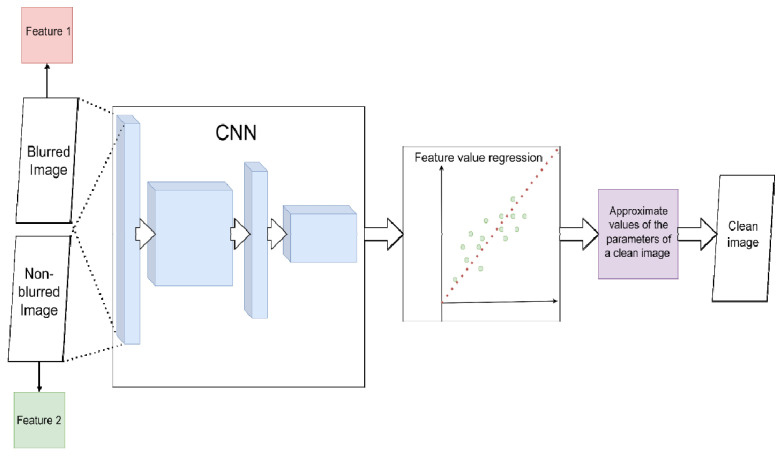
Scheme for image deconvolution with image-to-image regression using CNN (without PSF estimation). CNN directly solves the regression problem with the spatial parameters of the images. The parameters of the blurred image (red) are approximated to the parameters of the non-blurred (green). The resulting estimate is used to restore a clean image.

**Figure 6 micromachines-12-01558-f006:**
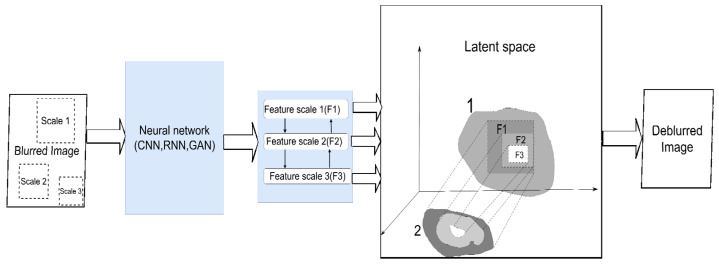
Principle of multiscale learning. A neural network of any suitable architecture extracts spatial features at different scales (F1, F2, F3) of the image. In various ways, smaller-scale features are used in conjunction with larger-scale features. This makes it possible to build an approximate explicit, spatially consistent clean image (2) from the latent representation of a clean picture (1).

**Figure 7 micromachines-12-01558-f007:**
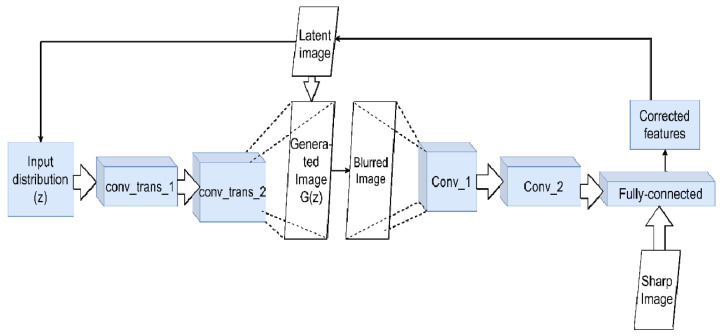
Using GAN for deconvolution. The generator network creates a false image G(z) from the initial distribution z (at first, it may be just noise). It is fed to the input of the network discriminator, which should distinguish it from the present. The discriminator network is trained to distinguish between blurred and non-blurred images, adjusting the feature values accordingly. These values form the control signal for the main generator. This signal will change G(z) step by step to approach the latent clean image iteratively.

**Figure 8 micromachines-12-01558-f008:**
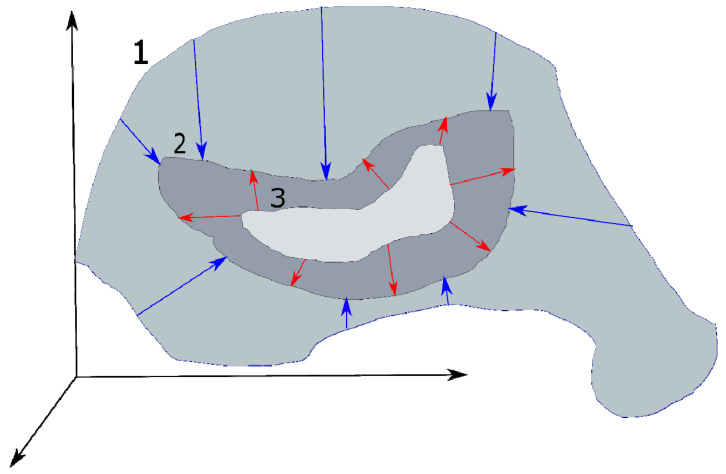
The principle of using the prior constraints. It narrows the infinite set of all possible values of the parameters of image 1 to the final one. Stronger prior constraints allow the narrowing of this set more precisely to the one corresponding to the true pure image 2 (shown by blue arrows). In turn, a more accurate and flexible model will iteratively refine its estimate of the clean image 3 (depicted by red arrows).

**Figure 10 micromachines-12-01558-f010:**
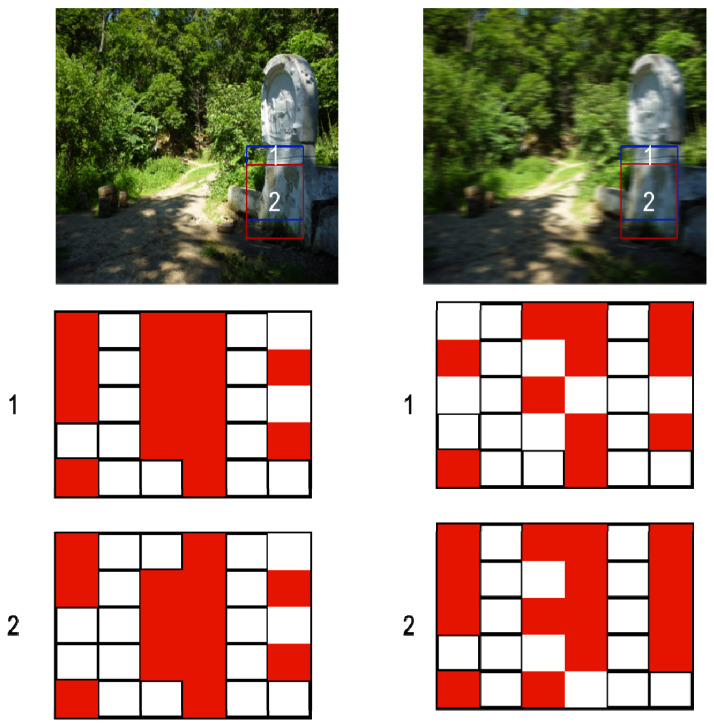
One of the examples of using sparse representation for image deconvolution (in particular, group sparse representation described in [117]). A clear image differs from a blurred image in that the overlapping areas will be very similar in a clear image. This means that a similar combination of atoms will describe them. The same spots in a blurred image will lose their similarity due to the influence of the blur kernel (especially a complex shape). The idea of patch similarity can be used as an initial idea. 1, 2 corresponds to different areas in the images and their representations.

**Table 1 micromachines-12-01558-t001:** A list of the deep learning software.

Software	Written in	Interface	Open Source	Auto Differentiation	Parallel Computing Support	Pre-Trained Model
Pytorch	C, C++, Fortran, CUDA, Python	Python, C++, Julia	Yes	Yes (dynamic graph)	Yes (CUDA, OpenMP, OpenCL)	Yes
Tensorflow 2	C++, CUDA, Python	Python (Keras), C/C++, Java, Go, JavaScript, R, Julia, Swift	Yes	Yes (dynamic graph)	Yes (CUDA)	Yes
MATLAB+ Deep Learning Toolbox	C, C++, Java, MATLAB.	MATLAB	No	Yes	Possible using additional modules (CUDA)	Yes
Deeplearning4j	C++, Java	Java, Scala, Clojure, Python (Keras), Kotlin	Yes	Yes	Yes (CUDA, OpenMP)	Yes
MXNet	C++	C++, Python, Julia, Matlab, JavaScript, Go, R, Scala, Perl, Clojure	Yes	Yes	Yes (CUDA, OpenMP)	Yes

**Table 2 micromachines-12-01558-t002:** A list of the most unified methods for testing and the maximum metric values they achieve.

Method	Dataset	Metrics	Metrics Value
CRCNet [63]	Levin	PSNR/SSIM	35.39/0.96
GSR-K [117]		PSNR	31.5
SASR [120]		PSNR/SSIM	30.91/0.92338
SRN-DeblurNet [64]	GoPro	PSNR/SSIM	30.1/0.9323
Deep Multiscale CNN for Dynamic Scene Deblurring [66]		PSNR/SSIM	29.08/0.9135
RCAN [149]		PSNR/SSIM	32.85/0.962
MSCAN-GoPro [150]		PSNR/SSIM	31.24/0.9423
SRN-DeblurNet	Koehler	PSNR/SSIM	26.80/0.8375
Deep Multiscale CNN for Dynamic Scene Deblurring		PSNR/SSIM	26.48/0.8079
RCAN		PSNR/SSIM	26.08/0.810

## Data Availability

Data sharing not applicable.

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
