# Peer review of "State-of-the-Art Approaches for Image Deconvolution Problems, including Modern Deep Learning Architectures"

_micromachines, 2021, doi:10.3390/mi12121558_

Round 1

Reviewer 1 Report

The authors presented an interesting topic, image deconvolution problems, and discuss the potential of DL in dealing with them. 

This important topic involves a real great deal of research, which is hard to summarize with reference to both theoretical and methodological aspects and also relatively to applications.

Being the proposed paper a review, I would invite the autora to substantially expand it. 

For instance, it would be useful to indicate and develop separately and more the theoretical modeling aspects and the computational ones, the latter related to algorithms and methods. 

The choice of discussing two major classes of DL seems to penalize other possible solutions (in part hybrid and in part different).

Problems related to training, testing and validation are not developed in-depth as they should. 

A list of software tools would be useful, and a dedicated table appreciated, including references and plus vs minus related to their use.

Applications domains should also be listed and report their specific impacts.

The figures appear to have a weak resolution, and should be improved.

Table 1 must indicate references too and improve in style.

Discussion and conclusions sections should benefit from introducing the above indicated elements and be adapted accordingly.  

Author Response

Being the proposed paper a review, I would invite the autora to substantially expand it. For instance, it would be useful to indicate and develop separately and more the theoretical modeling aspects and the computational ones, the latter related to algorithms and methods. 

We have substantially expanded the review in many parts and hope it has fixed this problem.

The choice of discussing two major classes of DL seems to penalize other possible solutions (in part hybrid and in part different).

We have added other classes of modern deep learning methods in the review, substantially extending it.

Problems related to training, testing and validation are not developed in-depth as they should. 

We have added separate part of article concerning training, testing and validation

A list of software tools would be useful, and a dedicated table appreciated, including references and plus vs minus related to their use.

We have added the table of major software tools used for implementation of deep learning algorithms for image processing.

Applications domains should also be listed and report their specific impacts.

We have added short discussion of application domains at the end of the review.

The figures appear to have a weak resolution, and should be improved.

We have improved the figure resolution in the new revision of the article.

Table 1 must indicate references too and improve in style.

We have added references to table and we have changed table presentation.

Discussion and conclusions sections should benefit from introducing the above indicated elements and be adapted accordingly.  

We have substantially extended and edited the discussion and conclusions sections of the article

Reviewer 2 Report

In the table 1 the identifier Levine  must be changed in Levin.

I suggest to add a reference for the datasets in table 1.

Author Response

Thanks for the nice opinion and the comments.

Point 1:

In the table 1 the identifier Levine  must be changed in Levin.

Response 1:

We have changed it throughout the text (now it is Table 2).

Point 2

I suggest to add a reference for the datasets in table 1.

Response2

We have added references (now it is table 2).

Reviewer 3 Report

The authors made a review of the state of the art approach for image deconvolution, there are several models which they are not mentioned.

Author Response

Point 1

The authors made a review of the state of the art approach for image deconvolution, there are several models which they are not mentioned.

Response 1

We have substantially extended the text and we hope now we're covering the abovementioned models too.

Round 2

Reviewer 1 Report

ok, the work has been sufficiently expanded and generally improved

This manuscript is a resubmission of an earlier submission. The following is a list of the peer review reports and author responses from that submission.

Round 1

Reviewer 1 Report

Authors present a review about the state of the art in image deconvolution focusing on classical, deep learning and optimization methods. The manuscript is well written and structured. They conclude that the manuscript constitutes a review of promising directions in microscopy, however none of the samples shown in the manuscript deals with the potential microscopy tools that are susceptible to deconvolution or even reported in deconvolution theory. For example, a paper showing a new optimization methodology in blind deconvolution has been recently reported in ophthalmoscopy imaging "Ávila, F.J.; Ares, J.; Marcellán, M.C.; Collados, M.V.; Remón, L. Iterative-Trained Semi-Blind Deconvolution Algorithm to Compensate Straylight in Retinal Images. J. Imaging 2021, 7, 73."

Minor revision:

Line 82: Figure 1 shows some examples of image degradation after convolution with complex PSF, details and image of the PSF must appear.

Line 181: Figure 2 states differences between blind and non-blind deconvolution. However the text is quite confusing for the reader as the process shown corresponds to convolution. 

Line 206: "DL uses multilayer neural networks" instead of "DL using
multilayer neural networks".

After correcting those minor revisions, I recommend the manuscript for publication.

Reviewer 2 Report

The paper is more of a review paper with not much of a novel contribution. Without proper comparison and results, it is hard to justify the modified loss function and the significance of the statements made. I would suggest restructuring the paper to present the significant contributions with substantial experimentations and comparisons.

Reviewer 3 Report

A clear rejection - see the comments of the referee
